# *E. coli* Phagelysate: A Primer to Enhance Nanoparticles and Drug Deliveries in Tumor

**DOI:** 10.3390/cancers15082315

**Published:** 2023-04-15

**Authors:** Ketevan Ghambashidze, Ramaz Chikhladze, Tamar Saladze, P. Jack Hoopes, Fridon Shubitidze

**Affiliations:** 1Department of Pathophysiology, Tbilisi State Medical University, Tbilisi 0177, Georgia; 2Department of Anatomic Pathology, Tbilisi State Medical University, Tbilisi 0177, Georgia; 3Geisel School of Medicine at Dartmouth College, Hanover, NH 03755, USA; 4Thayer School of Engineering at Dartmouth College, Hanover, NH 03755, USA

**Keywords:** tumor microenvironment, immunotherapy, *E. coli* phagelysate, tumor-associated macrophages, magnetic nanoparticle hyperthermia, cancer

## Abstract

**Simple Summary:**

The tumor microenvironment (TME) has a vital role in cancer development, progression, and metastasis. It can shape immune infiltration and activation within tumors, including the role of macrophages, tumor effector cells, and the uptake of therapeutic drugs or nanoparticles. In this manuscript, we report on the feasibility and preliminary efficacy of administering *E. coli* phagelysate (EcPHL) as a primer to modify the TME, including the enhanced uptake of magnetic nanoparticles (mNP) by tumor-associated macrophages (TAM) and tumor cells.

**Abstract:**

The tumor microenvironment (TME), where cancer cells reside, plays a crucial role in cancer progression and metastasis. It maintains an immunosuppressive state in many tumors and regulates the differentiation of precursor monocytes into M1 (anti-tumor)- and M2 (pro-tumor)-polarized macrophages, and greatly reduces anticancer drug and nanoparticle delivery. As a result, the effectiveness of recently developed chemo- and/or nanotechnology-mediated immune and magnetic nanoparticle hyperthermia (mNPH) therapies is inhibited significantly. One of the ways to overcome this limitation is to use *E. coli* phagelysate as a primer to modify the tumor microenvironment by switching tumor-associated M2 macrophages to anti-tumor M1 macrophages, and initiate the infiltration of tumor-associated macrophages (TAMs). Recently, bacteriophages and phage-induced lysed bacteria (bacterial phagelysates—BPLs) have been shown to be capable of modifying the tumor-associated environment. Phage/BPL-coated proteins tend to elicit strong anti-tumor responses from the innate immune system, prompting phagocytosis and cytokine release. It has also been reported that the microenvironments of bacteriophage- and BPL-treated tumors facilitate the conversion of M2-polarized TAMS to a more M1-polarized (tumoricidal) environment post-phage treatment. This paper demonstrates the feasibility and enhanced efficacy of combining *E. coli* phagelysate (EcPHL) and mNPH, a promising technology for treating cancers, in a rodent model. Specifically, we illustrate the EcPHL vaccination effect on the TME and mNP distribution in Ehrlich adenocarcinoma tumors by providing the tumor growth dynamics and histology (H&E and Prussian blue) distribution of mNP in tumor and normal tissue.

## 1. Introduction

Cancer is one of the most difficult diseases to cure. In 2022, the American Cancer Society (ACS) estimated there to be approximately 5250 new cancer cases each day, which totals about 1,918,030 cases in a year [1]. A total of 32% (609,360) of these total cancer cases are projected to be fatal [1]. Among tumor types, deep-seated cancers, such as lung, pancreatic, prostate, colorectal esophagus, and liver cancers, account for more than 50% (314,020) of cancer deaths in the USA [1]. The tumor microenvironment (TME), which surrounds a tumor, plays an integral role in cancer development, progression, and metastasis. The TME encompasses immune cells, extracellular matrix, blood vessels, and stromal and fibroblast cells [2,3]. The TME changes dynamically due to mutual interactions between the tumor and its associated microenvironment. Each individual tumor can contain various amounts of different cell types in the TME, but different cancer types tend to have overarching similarities [4]. For example, the microenvironment of many cancer types is very fibrotic or stiff. As a result, this greatly reduces drug (and nanoparticle) delivery to the tumor. To overcome these problems and drastically improve cancer survival rates, a new approach is needed to maneuver the crosstalk between the tumor and its microenvironment.

Recently, bacteriophages and phage-induced lysed bacteria (bacterial phagelysates—BPLs) have been shown to be capable of modifying the TME [5,6,7,8]. Phage/BPL-coated proteins and DNA tend to elicit strong anti-tumor responses from the innate immune system, prompting phagocytosis and cytokine release. It has been reported that the microenvironment of bacteriophage- and BPL-treated tumors switched from M2-polarized to a more M1-polarized environment post-phage treatment. Additional studies found that the administration of bacteriophage/BPL leads to an increase in neutrophilic granulocytes around the site of cancer. Most tumor TAM populations are largely composed of M2 pro-tumoral macrophages, which suppress immune activation in the TME and Th2 cytokine anti-inflammatory responses. Both M2 TAMs and anti-inflammatory cytokines (IL-10 and TGF-β) produce an inhibitory effect on cytotoxic CD8+ T cells, thereby facilitating tumor growth. M1 macrophages, in contrast, in response to bacterial stimuli and/or Th1 cytokines, become highly active phagocytes. When activated by lipopolysaccharide (LPS) and/or Th1 cytokine IFN-γ, M1 macrophages produce pro-inflammatory cytokines (TNFα, IL-12, IL-1β, IL-2, and IL-6), reactive oxygen species (ROS), nitric oxide (NO), and active anti-tumor functions. Our studies show that EcPHL (delivered on days 3 and 14 after tumor inoculation) modulates anticancer immune responses and delays very rapid murine Ehrlich adenocarcinoma growth in mice between 9 days and 17 days after implementing EAC.

In addition, magnetic nanoparticle hyperthermia (mNPH) is recognized as a promising cancer treatment technique that can be used alone or in combination with chemotherapy, radiation therapy, or immunotherapy for the treatment of a variety of cancers. In mNPH, magnetic nanoparticles are delivered to the tumor, either via direct tumor injection or intravenously. Once the mNP are in place, they are activated via an applied external alternating magnetic field (AMF), which causes the nanoparticles to heat, resulting in localized heating at the tumor and subsequent tumor cell cytotoxicity. Studies have shown hyperthermia to be effective for modifying the TAM and producing anti-tumor immune responses [9]. However, the main challenges, that is, the non-specific mNP uptake and activation/heating and the inability to get therapeutic levels of mNP to the target, remain a large deterrent. One way to address these challenges is by stimulating immune cells to reprogram the solid TME.

This paper introduces EcPHL as a primer to enhance mNP uptake in tumors by controlling TAM polarization. First, the EcPHL is delivered to the tumor via a systemic injection. The delivered EcPHL, which is a phage-based immune modulator, converts M2 phenotype TAM to M1 phenotype TAM. Then, the mNP are administered to the tumor via a local injection. Third, the delivered mNP are taken by the M1 phenotype TAM and tumor cells, which in return enhances, selectively, mNP tumor uptake at a level sufficient to result in successful tumor-specific hyperthermia treatment in the tumor and tumor bed.

## 2. Materials and Methods

This section summarizes the animals, cancer cell model, *E. coli* phagelysate modulator, and mNP, as well as describes histological and statistical methods that are used in this study.

### 2.1. Animals

Eight- to ten-week-old male BALB/c mice, weighing 20–25 g were purchased from the vivarium of Aleksandre Natishvili Institute of Morphology, Tbilisi, Georgia (https://www.tsu.ge/en, accessed on 10 February 2022). Animal use was reviewed and approved by the Tbilisi State Medical University Animal Welfare and Use Ethics Committee. The mice were housed eight per cage and were given standard laboratory chow and water. They were allowed to acclimate to the laboratory conditions for one week before the experiment began.

### 2.2. Cell Lines

Mouse tumors were propagated from Ehrlich ascites carcinoma cells, also known as EAC cells. EAC cells are derived from epithelial cells and are highly malignant. In cell culture, the cells are round or oval and have a large nucleus and a small amount of cytoplasm. The EAC has been extensively utilized for studies on TME, TAMs, immunology, chemotherapy, and for assessing the efficacy of various anticancer drugs.

In subsequent studies, we used the EAC cells provided by the Kavetsky Institute of Experimental Pathology, Oncology and Radiobiology of National Academy of Sciences of Ukraine, Department of Experimental Cell Systems (https://iepor.org.ua and www.onconet.kiev.ua accessed on 3 February 2022). The provided cells resemble human tumors in several key ways. For example, like human tumors, EAC cells are undifferentiated, meaning that they have not fully matured and do not have a specific tissue type. As a result, the cells grow and spread quickly, making them useful for studying the growth and spread of cancer and for monitoring mNP and anticancer drug efficacy. Additionally, EAC cells have a high transplantable capability, meaning that they can be easily transplanted from one animal to another. This makes them useful for studying the mechanisms of cancer in living organisms. For propagation purposes, five million EAC cells were injected intraperitoneally on a biweekly basis. Cells were counted with the use of hemocytometer. The experiment began one week following subcutaneous transplantation of EAC, at which point the average tumor volume was 0.3 cm^3^.

### 2.3. Tumor Propagation

Under brief ether anesthesia, each mouse was inoculated subcutaneously in the right flank with (2 × 10^6^ cells/20 g body weight) freshly drawn from a donor mouse. The tumor cells were counted using a hemocytometer prior to their subcutaneous inoculation. The viability of the EAC cells was 98%, as judged by trypan blue exclusion assay.

Cancer-bearing mice were randomly divided into four groups:Group I: control group, no treatment.Group II: treatment with anti-tumor immunomodulator—EcPHL.Group III: treatment with Fe-containing magnetic nanoparticles—mNP.Group IV: combined treatment—EcPHL plus mNP.

In an additional study of the lifespan and percentage of survival, an additional group of 15 untreated mice and 15 treated mice were used.

### 2.4. Tumor Size Measurements

The growth of the tumors generated from EAC cells was monitored using a Vernier caliper. The volume of the tumor was calculated using the formula V = length × width × 0.52, which considers the ellipsoidal shape of the tumor. This assessment was performed every third day to track the growth of the tumors over time.

The treated and control groups’ tumor growth inhibition percent was calculated using the formula: %TGI = (MTVcontrol − MTVtreated)/MTVcontrol × 100. Tumor growth inhibition ≥ 50% was considered meaningful.

Cancer growth average rate (mm^3^/day) was calculated using the formula: (V2 − V1)/n (n—number of days).

### 2.5. Biopreparation

The biopreparation of EcPHL, also known as *E. coli* phagelysate, was performed using an immunomodulatory preparation made from bacteriophages (viruses that infect bacteria). It was produced by the George Eliava Institute of Bacteriophage, Microbiology and Virology in Tbilisi, Georgia, http://eliava-institute.org, accessed on 12 January 2022. The final EcPHL had a titer of 2 × 10^10^, fr.8 concentration of the active ingredient in the preparation. This product is commonly used in cancer research to study the potential immune-modulating effects of bacteriophages on cancer cells.

The phagelysates were prepared in synthetic M9 medium supplemented with 0.1% yeast extract, inoculated with an overnight suspension of *E. coli* (C) 600 with a final concentration of 1–5 × 10^7^ cfu/mL, and incubated with shaking at 37 °C [10,11]. The Un phage [12] was added to the culture at the exponential growth phase (OD = 0.5 at 660 nm) at a phage–bacteria ratio of 1:10 and incubated with shaking for 4–5 h at 32 °C. The resulting lysate was kept overnight at 4 °C, then centrifuged at 4000× *g* for 20 min, and filtered through a 0.8 µm pore size cellulose nitrate membrane filter (Whatman International, Maidstone, UK). The filtrate was characterized by measuring the phage titer, optical density at 660 nm, and protein concentration using the Bradford method [13]. The endotoxin content in all phagelysate preparations was evaluated using the gel-clot test and Limulus amebocyte lysate (LAL) assay (Associates of Cape Cod, Inc., East Falmouth, MA, USA) according to the manufacturer’s instructions. Endotoxins were detected in all tested preparations, with varying levels exceeding 0.03 EU. The phagelysates were also tested for safety and pyrogenic effects on healthy white and black mice. A mild pyrogenic effect (slight increase in temperature) within 2 h after inoculation of phagelysates as well as some behavioral changes (primarily dizziness) were observed with normalization within a few hours. No deaths were reported. For animal trials, phagelysates of *E. coli* C-Un with an average phage titer of 1 × 10^10^ pfu/mL and protein concentration of 0.12 mg/mL were used.

### 2.6. Immunomodulation

In these studies, the EcPHL was administered to mice by intraperitoneal injection. The injections were given 3 times, with 3 days in between each injection. The injections were started after the tumors reached an average size of 300 mm^3^.

### 2.7. Magnetic Nanoparticles

Magnetic nanoparticles developed at Dartmouth College, Hanover, NH, USA, were used in this study. The magnetic nanoparticles (mNP) were composed of 2–5 nm individual crystals of γ-Fe_2_O_3_, embedded with saccharide chains within their crystalline structure. These crystals aggregate to form flower-like structures that are 20–40 nm in size, with a hydrodynamic diameter of 110–120 nm. The mNP are capable of forming stable colloidal solutions in water that can last for more than 12 months. These mNP exhibit no hysteresis when subjected to a quasi-static magnetic field and are able to generate significant heat, even at low field strengths of just 100 Oe, within a frequency range of 99–164 kHz, and their saturation magnetization, remanence, and coercivity are 1.1 emu/g, 0.007 emu/g, and 30 µT, (0.3 G), respectively. More detailed information about Dartmouth mNP shapes, sizes, magnetic properties, and heating mechanisms can be found in our previously published manuscripts [14,15]. Dartmouth nNPs produce therapeutic levels of SAR at low AMF strength, which makes them advantageous for deep-seated tumor mNP hyperthermia cancer therapy, where high field strengths are not practical using an external alternating magnetic field.

### 2.8. Histological Estimation

To determine the effects of the treatments, the mice were sacrificed after 24 days, i.e., 17 days after the EAC cell inoculation, and tumor tissue samples were taken for histological investigation. The tissue samples were taken from each group on the same day, but samples from the mice in Groups III and IV were taken at different time points after the administration of mNP. Specifically, tissue samples were taken 3, 6, 12, and 24 h after mNP administration to track the distribution of the nanoparticles in tumor tissue over time. Once the tissue samples were collected, they were fixed in formalin, embedded in paraffin, cut at 4 microns, and stained with hematoxylin and eosin (H&E) or with Perl’s Prussian blue stain. Each cut section was examined using a Leica DM6B microscope and Motic Easy Scan Pro scanner (ultrahigh NA APO 20X (NA 0.75), up to 40X magnification and 0.25 μm/pixel resolution). The mNP spread was analyzed for each cut by measuring distances between farther points for two different directions. 

### 2.9. Statistical Analysis

The data were assessed statistically using IBM SPSS statistics to determine the significant differences between the control group and the treated group. The criterion for significance, in this case, was set to *p* < 0.05.

## 3. Results

In this section, we present the tumor growth kinetics, tumor histology, and mNP distribution dynamics. All subsequent studies are performed for four groups. First, the tumor growth is compared between the control and EcPHL anti-tumor immunomodulator-vaccinated groups. Next, the histology analyses are presented for all four groups, and finally, the mNP distributions are presented in the control and EcPHL-treated mice after different time periods, and the applicability of the EcPHL immunomodulator vaccine is assessed for shifting the TME pro-tumor functions toward anti-tumor functions for enhancing mNP uptake in and around the TME to enhance mNP hyperthermia efficacy.

### 3.1. Tumor Growth

The results of the experiment indicate that anti-tumor immunomodulation with EcPHL can significantly inhibit tumor growth. On post-EAC implant days 9, 12, 15, and 16, mean tumor volume (MTV) was reduced by 2.2x, 2.6x, 2.3x, and 1.9x, respectively, compared to the control group of mice (*p* < 0.002, *p* < 0.001, *p* < 0.001, and *p* < 0.001, respectively). On post-EAC implant days 9, 15, and 16, the tumor growth average rate was decreased by 3x, 1.25x, and 2x (*p* < 0.002, *p* < 0.001, and *p* < 0.001, respectively), and on post-EAC implant days 9, 12, 15, and 16, the tumor growth inhibition percentage (%TGI) was 54%, 64%, 56%, and 46% (*p* < 0.002, *p* < 0.001, *p* < 0.001, and *p* < 0.001, respectively). These results suggest that EcPHL has a powerful inhibitory effect on tumor growth. The results depicted in Figure 1 show that the EcPHL immunomodulator slows down the tumor growth rate significantly, particularly on the 12th day.

To further demonstrate the efficacy of the EcPHL immune modulator, Figure 2 shows comparisons between the tumor tissue samples for the EcPHL-treated and control (untreated) mice. The visual comparisons clearly show the difference in tumor tissue size between treated and untreated Ehrlich-carcinoma-bearing mice.

### 3.2. Histological Analysis of Mice Tissue

#### 3.2.1. Control: Group I

A subgroup of animals was sacrificed on the 24th day of post-tumor initiation. The tumor tissues were taken and prepared for histological examinations. The histological samples were routinely processed and cut as 4-micron thin sections. However, for understanding the dynamics of mNP distributions in and around the tumor, the tissue samples were taken at different 6 h and 24 h times after the MNP administration for Groups III and IV. A histological analysis of Group I showed abundant tumor/neoplastic cells with modest connective tissue, stroma, and moderate multifocal aggregates of inflammatory cells, as well as multifocal necrosis, as seen in Figure 3.

Figure 4 shows that the carcinoma tumor infiltrates the skin, and there are significant inflammation and desmoplasia. The cells have a large cytoplasm, indicating that the parenchyma predominantly consisted of atypical, anaplastic cells with abundant basophilic cytoplasm. In addition, within the tumor necrotic zones, coagulative and liquefactive tissues can be observed, and neutrophils are accumulated in and around the infected cells.

#### 3.2.2. Group II: Mice Vaccinated with EcPHL Immune Modulator

In this group, there is a morphologically clear mixing of large macrophages and tumor cells. In some sections, macrophages dominate (Figure 5a,b). There is also multifocal necrosis (Figure 5a).

These results clearly show that there are a significant number of macrophages in and around the tumor. These large numbers of TAMs are a type of immune cell, which are part of the tumor microenvironment. There are two phenotypes: M1- and M2-polarized TAMs. The ratio between the polarized M2 and M1 determines TAM efficacy during combating cancer cells. Namely, M1 macrophages have pro-inflammatory responses and play a key role in eliminating tumor cells. M2 macrophages, on the other hand, are alternatively activated macrophages that are involved in anti-inflammatory responses and promote cancer cell growth. The results show that the EcPHL activates the TAMs; however, the ratio of M2 and M1 macrophages is not clear.

#### 3.2.3. Group III: EcPHL-Untreated Mice Administered with mNP via Local Intratumor Injection

In this group of mice, mNP (5 mg/per gram of tumor) were injected into the tumor via local injection, and histology samples were taken at 6 and 24 h after injection. Figure 6 shows the histology of samples taken at 6 h and 24 h after mNP administration. The results show the aggregation of the exogenous mNP-containing pigment, as well as macrophage concentrations in the injection area.

#### 3.2.4. Group IV: EcPHL- and mNP-Treated Mice

The EcPHL-treated mice were administered with the mNP via intratumor injection. The tumor tissue samples were taken at 6 and 24 h after injection for histology. Figure 7 and Figure 8 show the history of EcPHL-treated tissue samples taken at 6 h and 24 h after mNP intratumor injection. The results show that the mNP-containing exogenous pigment in the form of cytoplasmic granules is significantly concentrated in the necrotic areas, macrophages, and fatty and connective tissue under the skin, and the number of mNP-carrying macrophages decreases gradually from the center of particle injection to the periphery. However, the areas of inflammatory infiltration and the concentration of mNP-containing cells are increased and more visible. Comparisons between the spread of mNP-carrying macrophages from the particle injection areas illustrate that mNP particles are spread across larger distances in the EcPHL-treated mice (Group IV; Figure 9) than in EcPLH-untreated mice samples (Figure 10), especially in the samples that were taken at 24 h after mNP intratumoral injection (Figure 6 and Figure 8).

The tissue sample shows iron-containing exogenous pigment accumulation in the necrotic areas, and fatty and connective tissue, intra- and extracellularly, as presented by the large amounts of concentrated masses, inflammatory infiltration, and iron-containing cells. The tissue sample demonstrates abundant mNP/iron-containing macrophages at the exterior boundary of the tumor, as seen in Figure 8.

Overall, a histological examination of tumor tissue samples indicates that EcPHL- and mNP-treated mice had greater numbers of mNP-carrying macrophages. These particles accumulated in the necrotic areas and fatty and connective tissue, extra- and intracellularly. The examinations show that there are more areas of inflammatory infiltration and a higher concentration of iron-containing cells, as well as iron-containing macrophages, at a greater distance from the mNP injection in the EcPHL-treated mice samples than in the non-vaccinated mice.

### 3.3. EcPHL Immunomodulator Treatment Effect

Finally, in addition to the four experimental groups, we assessed EcPHL immunomodulator cancer treatment efficacy by monitoring the lifespan of the subset control and EcPHL-treated mice. A total of 15 control and 15 EcPHL-treated mice were monitored, and the percentage of surviving mice was calculated. The results are depicted in Figure 11. The comparisons between lifespans for treated and untreated mice show that the EcPHL-treated mice had twice the lifespan of the control mice, and the percentage of surviving mice was three times more compared to the untreated control group mice. By the 65th day of Ehrlich carcinoma growth, all of the control group mice were deceased, while 60% of the mice were still alive in the EcPHL-treated group.

## 4. Discussion

Ehrlich carcinoma has been used as a model system to study cancer biology, tumor immunology, and anti-tumor therapies. The model replicates many types of human cancers, including pancreatic, gastric, colorectal, liver, ovarian, lung, and lymph node cancers [16,17,18,19,20,21,22,23,24]. It is a highly malignant tumor that is composed of undifferentiated cells and can metastasize to various organs [20,21,22,23,24]. This model has been used to evaluate the efficacy of various chemotherapeutic agents, nanoparticles, and immunotherapies, including monoclonal antibodies, immunomodulators, immune checkpoint inhibitors, and adoptive cell therapies, as well as the TME [25,26,27,28,29,30,31,32].

Studies demonstrate that the TME maintains an immunosuppressive state and malignant tumors recruit pro-tumor M2 TAMs to the TME [24,33]. M2-polarized macrophages are known for their anti-inflammatory and pro-tumorigenic properties, and they are characterized by the expression of markers such as CD206, Arginase-1 (Arg-1), and Ym1/2. In the TME, M2-polarized macrophages promote tumor growth and metastasis through various mechanisms. They can secrete factors such as transforming growth factor-beta (TGF-β), interleukin-10 (IL-10), and vascular endothelial growth factor-A (VEGF-A), which promote angiogenesis. M2-polarized macrophages also inhibit anti-tumor immune cells, such as cytotoxic T cells and natural killer (NK) cells [24,34,35,36,37].

Switching pro-tumor M2-polarized macrophages toward anti-tumor M1-polarized phenotype macrophages in the TME has been considered a potential therapeutic strategy for inhibiting tumor growth and metastasis [24,34,38]. One of the approaches is to use immunomodulatory agents, such as EcPHL. The EcPHL active components (e.g., bacterial LPS, unmethylated CpG oligodeoxynucleotide (CpGODN) fragments, and phage-coated proteins) stimulate the production of pro-inflammatory cytokines, such as interferon-gamma (IFN-γ), tumor necrosis factor-alpha (TNF-α), IL-12, TNF-α, IL-6, and nitric oxide (NO) [31,39,40,41]. Unmethylated CpG motifs directly stimulate B cells and T cells. In turn, they promote antibody production and cytotoxic T lymphocyte (CTL)-cell-mediated immune responses [42].

Our studies in [31,43,44] revealed that the EcPHL immune modulator produces strong Th1-type responses in the Ehrlich-carcinoma-bearing mice tumor tissue. Furthermore, the increased numbers and functional activities of NK1, CD4+, and CD8+ cells were shown. Namely, increased secretion of IL-12 and IFN-γ (along with the further decrease in TGF-ß, CD4CD25, and CD25Foxp3) was observed in the EcPHL-treated mice in comparison to the control.

The studies showed that:(a)EcPHL exhibits strong anticancer immunomodulatory properties affecting the TME. The EcPHL efficacy was strongest on the 9th and 12th days of Ehrlich carcinoma growth, after the first and second EcPHL vaccinations, which could be related to the increased ratio of M1/M2 macrophages in the TME [45,46].(b)Histological analyses of the EcPHL-treated mice tumor tissue samples (see Figure 5, Figure 6, Figure 7 and Figure 8) show an increased number of macrophages (supposedly M1 phenotype), which, in turn, release cytokines (e.g., IL-12, TFN-α, and IFN-γ), stimulate further immune system responses, specifically in and around the tumor lesions, and promote tumor cell killing. The observed EcPHL efficacy and the effect of the macrophages on Ehrlich carcinoma are consistent with the view that M1 macrophages possess anti-tumor properties [24,47,48,49].(c)Although an increase in the M1/M2 ratio in the TME should provide better clinical outcomes in cancer [46], the balance between M1 and M2 macrophage polarization in the TME is influenced by a range of factors, including cytokines, chemokines, and other signaling molecules produced by tumor cells. Even though, at present, we cannot determine which factors of the exogenous EcPHL vaccine activated M1 macrophages and how they impacted Ehrlich carcinoma growth, we hypothesize that EcPHL overdose can reduce the ratio of M1/M2 macrophage polarization in the TME, and as a result, it can decrease EcPHL vaccination efficacy.(d)EcPHL improves mNP distributions and the spread radius in the tumor tissue from the local inoculation point in comparison to mice tumor tissue without immunomodulation. The mNP were observed in the necrotic areas and fatty and connective tissues, as well as in the TAMs, both intra- and extracellularly. Intracellular iron-containing exogenous pigments were visible in the cytoplasmic granule. A large number of mNP were observed in inflammatory infiltrates, necrotic areas, and macrophages.

## 5. Conclusions

In this paper, the applicability of the EcPHL was assessed and demonstrated for modifying the TME/TAM and for enhancing mNP delivery in and around tumor cells. The studies demonstrate that the EcPHL is a strong immunomodulator capable of modifying the TME by activating macrophages and supposedly switching M2 phenotype to M1 phenotype macrophages. The activated macrophages gravitate to hypoxic regions, and as a result, they are able to clear damaged cells and tissue and promote mNP movement into tumor tissue. Although these studies are focused on enhancing mNP delivery in the tumor tissue in order to improve mNPH cancer therapy, one can engineer mNP to bind to the membrane of cancer cells for targeted delivery of therapeutic agents. Once bound, the mNP can be internalized into the cell through endocytosis, and the therapeutic agents can be released within the cytoplasm, where they can deliver desirable biological effects. In addition, the internalized mNP can also be used for biosensing applications to detect TAMs or the molecules that they secrete, such as cytokines or chemokines. In future studies, we plan to further understand the underlying mechanism behind how the EcPHL immune modulator interacts with the TME and use this knowledge to further improve mNP delivery in the TME for mNPH cancer therapy. Namely, we aim to map M1 and M2 phenotype macrophages in the TME and to activate mNP in the tumor using external alternating electromagnetic fields.

## Figures and Tables

**Figure 1 cancers-15-02315-f001:**
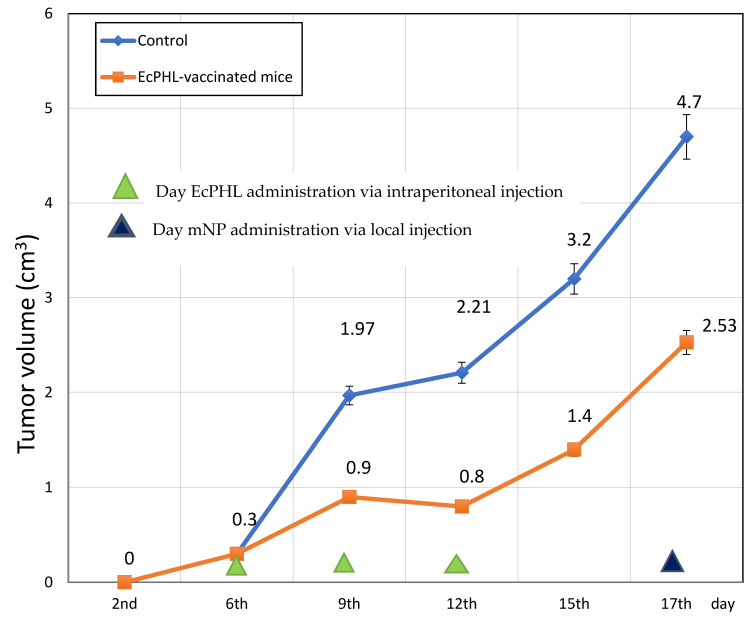
Tumor volume (cm^3^) in the Group I mice (control, untreated) and the Group II EcPHL-vaccinated mice.

**Figure 2 cancers-15-02315-f002:**
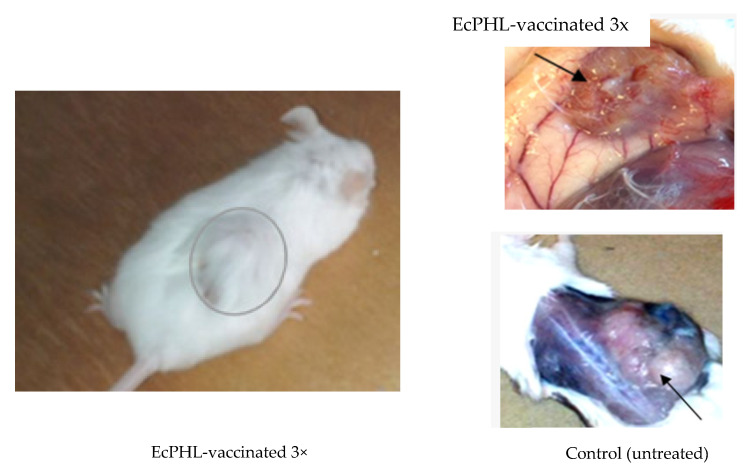
EcPHL-treated and control (untreated) mice and their corresponding tissues.

**Figure 3 cancers-15-02315-f003:**
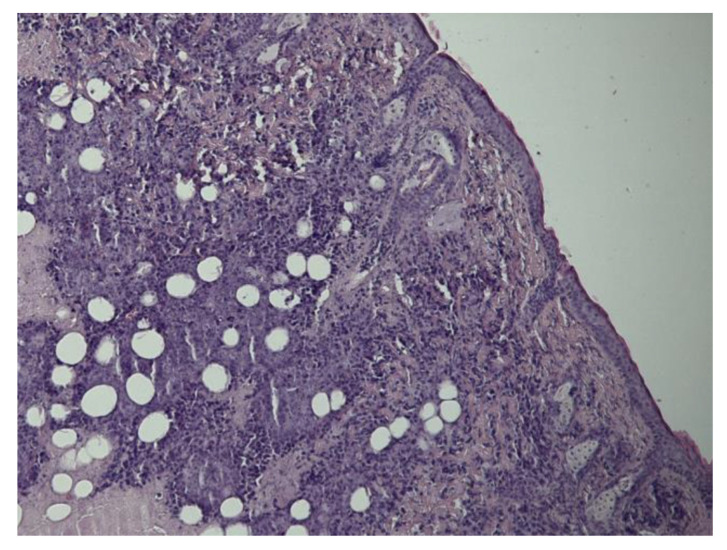
This photomicrograph (H&E stain, 100×) shows the carcinoma tumor infiltrating the skin. There is significant inflammation, necrosis and desmoplasia.

**Figure 4 cancers-15-02315-f004:**
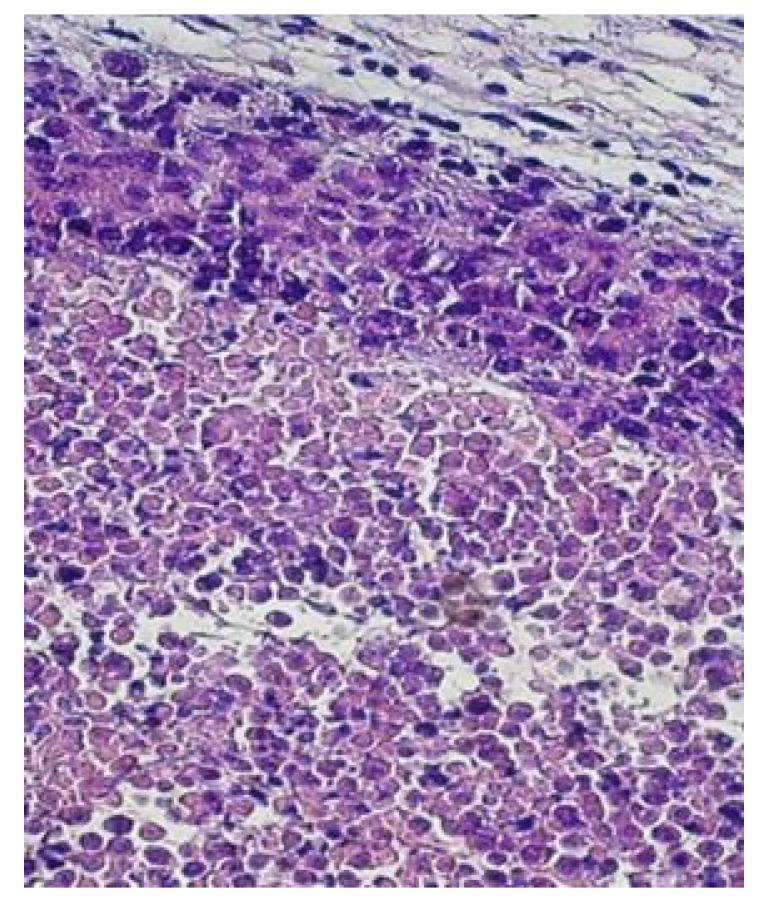
Histopathological demonstration of the outer margin of the tumor (magnification ×200). There is a clear boundary between the tumor and surrounding normal tissue. The outer part of the tumor appears to contain numerous macrophages.

**Figure 5 cancers-15-02315-f005:**
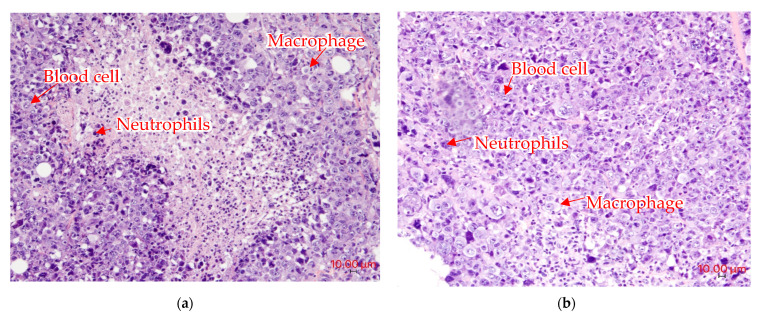
Histopathological demonstration of significant tumor necrosis (Ehrlich-carcinoma. Group II mouse) following EcPHL treatment. Necrotic areas show intermixing of dead and dying tumor cells and inflammation. (**a**)—(H&E stain, magnification ×100); (**b**)—(H&E stain, magnification ×200).

**Figure 6 cancers-15-02315-f006:**
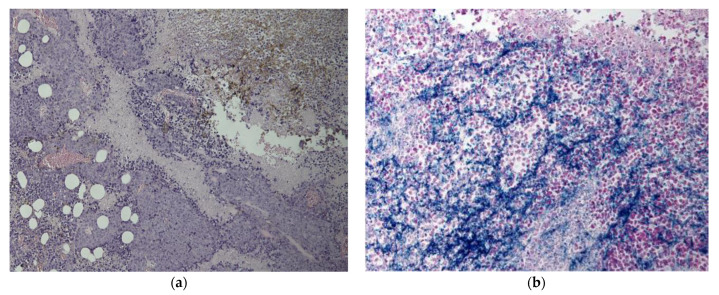
Histological demonstration of significant tumor tissue necrosis (Ehrlich carcinoma) and mNP accumulation in macrophages 24 h after mNP intratumor injection. (**a**)—(H&E ×100); (**b**)—(Perl’s Prussian blue ×100). Group III mouse.

**Figure 7 cancers-15-02315-f007:**
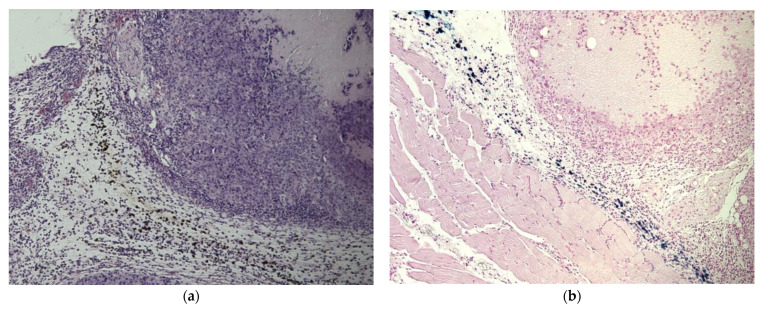
These histological sections demonstrate significant mNP accumulation in the peritumoral (**a**,**b**) and the intratumoral region (**a**) following EcPHL treatment and mNP administration (6 h prior to sampling). H&E staining (**a**), Perl’s Prussian blue staining (**b**). Magnification: ×100.

**Figure 8 cancers-15-02315-f008:**
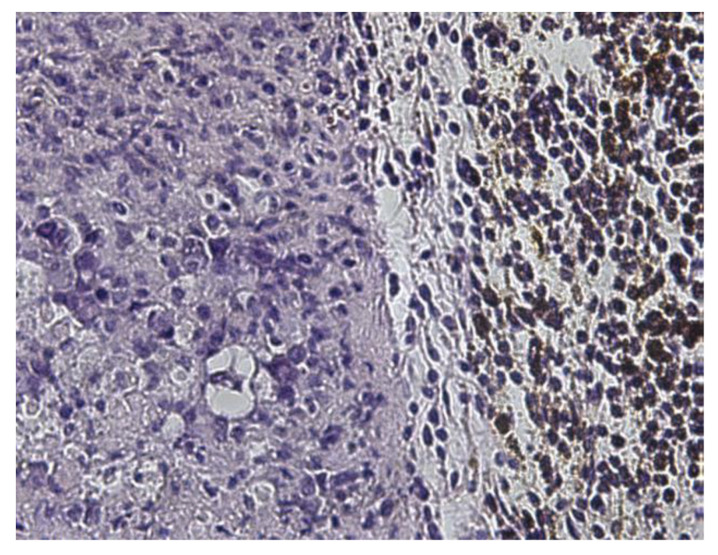
This histological section demonstrates significant mNP accumulation in per-tumoral macrophages following EcPHL vaccination and mNP administration 24 h post mNP intratumor injection. H&E staining, Magnification: ×400.

**Figure 9 cancers-15-02315-f009:**
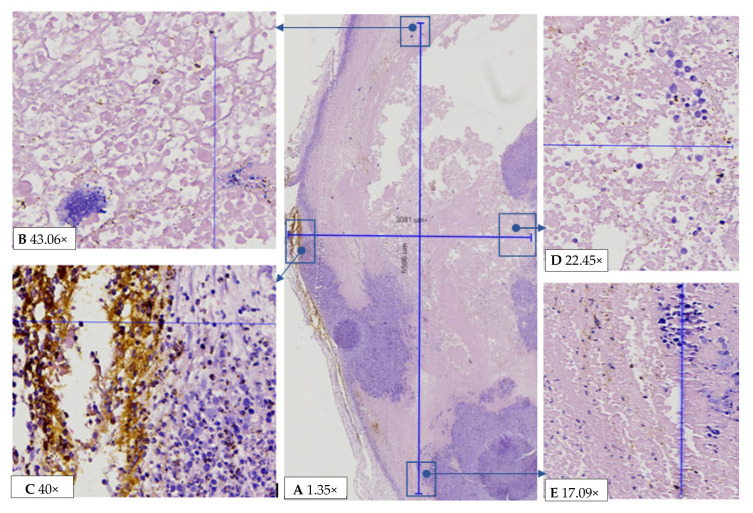
Significant tumor necrosis (**A**–**E**) and marked accumulation of mNPs and inflammatory cells at the tumor normal tissue interface. The tissue sample was acquired 6 h following intratumoral mNP delivery. Group IV mouse. H&E staining. (**A**) magnification: length, 5.596 μm; width, 3.081 μm.

**Figure 10 cancers-15-02315-f010:**
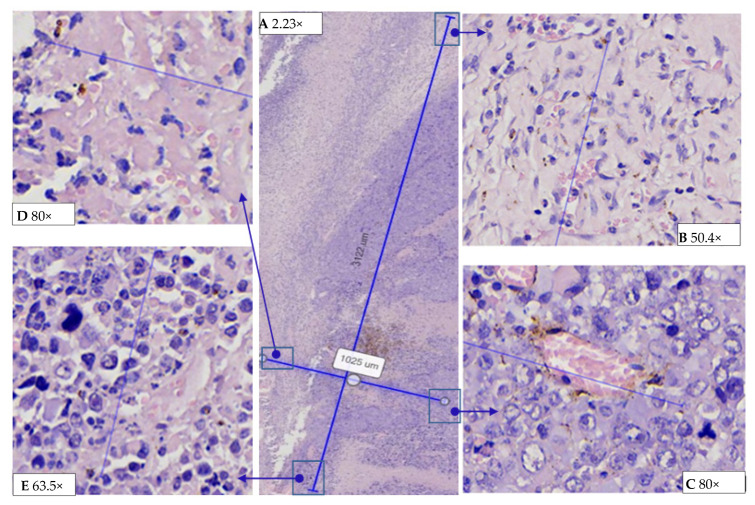
Photomicrographs demonstrate histopathological effect in a tumor following EcPHL vaccination and intratumoral mNP injection (6 h prior sampling). There is significant tumor necrosis (**A**,**D**) as well as early fibrous infiltration (**B**) and macrophage -tumor cell intermixing (**C**,**E**). (**C**) demonstrates the accumulation of mNP around a tumor blood vessel. Group III mouse. H&E staining. (**A**) magnification: length, 3.122 μm; width, 1.025 μm.

**Figure 11 cancers-15-02315-f011:**
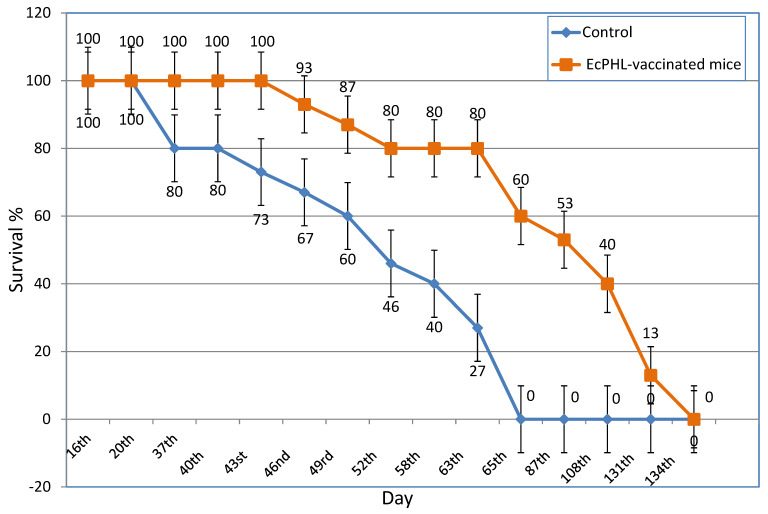
Lifespan and percent of surviving untreated (15 mice) and EcPHL-vaccinated (15 mice) cancer-bearing mice.

## Data Availability

The data presented in this study are available upon request.

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
