# Peer review of "E. coli Phagelysate: A Primer to Enhance Nanoparticles and Drug Deliveries in Tumor"

_cancers, 2023, doi:10.3390/cancers15082315_

Round 1
Reviewer 1 Report
The authors have focused on the study of tumor response modification to achieve greater treatment efficiency in the progression of metastasis to shrink or eliminate the tumor. Thus they study the la administering E.coli phagelysate (EcPHL), as a primer to modify the TME, including the enhanced uptake of magnetic nanoparticles (mNP) by tumor associated macrophages (TAM) and tumor cells
Materials
Cell lines, do not indicate culture medium, incubation time or suitable passages. The description that is made would be of more interest in the results section.
Tumor Propagation, no mention is made of how the cells were prepared, or how the number of cells was determined.
Biopreparation, there is no detail on how to perform it, the instrumentation is insisted on. Perhaps some reference could be introduced
In each group of mice subjected to treatments, how many are part of each group? 15?
Discussion
Authors should reflect comments on their histological studies.
General comments
Fig 1 and 2 do not give information
Fig 3 lacks names of the axes. In figure 3 the units of size in volume (cm-3) should be written properly.
In line 206 paragraph 266-278, abbreviations are missing. In line 340, "-" should be added in references 33-37 or separate them "[ ]".
Figures 5 and 6 contain letters or numbers that the authors do not reflect in the figure caption or in the main text. Likewise Fig 9, Fig 10.
In Fig 10, reference is made to "um", perhaps they refer to "µm".
Figure 6 does not reflect the magnifications.
Care should be taken in the way of writing units of measurement in the text and figures, especially in the volumetric ones.
Care should be taken when writing units of measurement in the text and figures, especially in volumetric units
Author Response
The authors have focused on the study of tumor response modification to achieve greater treatment efficiency in the progression of metastasis to shrink or eliminate the tumor. Thus they study the la administering E.coli phagelysate (EcPHL), as a primer to modify the TME, including the enhanced uptake of magnetic nanoparticles (mNP) by tumor associated macrophages (TAM) and tumor cells.
We thank the reviewer for comments.
Materials
Cell lines, do not indicate culture medium, incubation time or suitable passages. The description that is made would be of more interest in the results section.
Additional materials have been added.
Tumor Propagation, no mention is made of how the cells were prepared, or how the number of cells was determined.
A paragraph describing the tumor cells preparion and propagation added.
Biopreparation, there is no detail on how to perform it, the instrumentation is insisted on. Perhaps some reference could be introduced
References added
In each group of mice subjected to treatments, how many are part of each group? 15?
Clarifications added.
Discussion
Authors should reflect comments on their histological studies.
Refence to Figures 5-8 has added.
General comments
Fig 1 and 2 do not give information
Deleted
Fig 3 lacks names of the axes. In figure 3 the units of size in volume (cm-3) should be written properly.
Corrected.
In line 206 paragraph 266-278, abbreviations are missing. In line 340, "-" should be added in references 33-37 or separate them "[ ]".
Corrected
Figures 5 and 6 contain letters or numbers that the authors do not reflect in the figure caption or in the main text. Likewise Fig 9, Fig 10.
Changed.
In Fig 10, reference is made to "um", perhaps they refer to "µm".
Yes, um means µm. We would like to keep um.
Figure 6 does not reflect the magnifications.
Added.
Care should be taken in the way of writing units of measurement in the text and figures, especially in the volumetric ones.
Corrected.
Care should be taken when writing units of measurement in the text and figures, especially in volumetric units
Updated
Reviewer 2 Report
Authors showed in this manuscript that the EcPHL was a strong immunomodulator capable of modifying the TME by activating macrophages; also, for enhancing mNP delivery in and around tumor cells. The paper has good structure and can be published after revision.
What are the limitations of the study and how could they be addressed in future research?
Can you discuss the potential clinical implications of the study's findings?
How do the results of this study compare to other studies in the field?
What are some potential future directions for research based on the findings of this study?
Author Response
Authors showed in this manuscript that the EcPHL was a strong immunomodulator capable of modifying the TME by activating macrophages; also, for enhancing mNP delivery in and around tumor cells. The paper has good structure and can be published after revision.
What are the limitations of the study and how could they be addressed in future research?
These are preliminary studies. Additional studies are needed to understand how EcPHL affect on TME, and when is the bets time to deliver mNPs and drugs. We need to use immune cells markers to quantify M2-M1 polarization changes, and understand how mNP mediated heat affect on the TAM.
Can you discuss the potential clinical implications of the study's findings?
Although, the mNP hyperthermia and EcPHL are well understood and studies there are still needs to further investigate how combine approach will affect TME. As next step of this investigation, before clinical testing, we believe that studies should be done on large animals.
How do the results of this study compare to other studies in the field?
The studies for each approach (EcPHL and mNP) presented here are reported by other groups in the literature. However, the proposed combine approach is novel, and could open a new platform drug and nano particles delivery in tumor cells.
What are some potential future directions for research based on the findings of this study?
Furter directions are: activating mNPs, understanding TAM M2-M1 polarization changes, reprogram TAM using EcPHL and mNP mediated heat
Reviewer 3 Report
Thank you for the opportunity to review the manuscript.
The authors have managed to present the data clearly, although there are some minor typing and formatting errors throughout. However, it is suggested to include a flowchart or diagram to summarize the method section especially, to ease understanding.
Although the authors have previously published on the Dartmouth NP, it would be beneficial to the readers if the authors can include some information on the nanoparticles, e.g the material used and type of the magnetic nanoparticles. The authors also did not mentioned on the setting used for the AMF treatment in the study.
Author Response
Thank you for the opportunity to review the manuscript.
We thank the reviewer for reading and commenting the paper. Your inputs and suggestions makes the paper stronger and better understandable for the readers.
The authors have managed to present the data clearly, although there are some minor typing and formatting errors throughout. However, it is suggested to include a flowchart or diagram to summarize the method section especially, to ease understanding.
Thanks for the comments. We have made updates, added paragraphs to clarify our points.
Although the authors have previously published on the Dartmouth NP, it would be beneficial to the readers if the authors can include some information on the nanoparticles, e.g the material used and type of the magnetic nanoparticles. The authors also did not mentioned on the setting used for the AMF treatment in the study.
Additional materials have added.
Round 2
Reviewer 2 Report
It can be published.